# Adaptation of Livestock to New Diets Using Feed Components without Competition with Human Edible Protein Sources—A Review of the Possibilities and Recommendations

**DOI:** 10.3390/ani11082293

**Published:** 2021-08-03

**Authors:** Marinus F. W. te Pas, Teun Veldkamp, Yvette de Haas, André Bannink, Esther D. Ellen

**Affiliations:** 1Animal Breeding and Genomics, Wageningen University and Research, Droevendaalsesteeg 1 (Building 107), 6708 PB Wageningen, The Netherlands; yvette.dehaas@wur.nl (Y.d.H.); esther.ellen@wur.nl (E.D.E.); 2Animal Nutrition, Wageningen University and Research, De Elst 1 (Building 122), 6708 WD Wageningen, The Netherlands; Teun.Veldkamp@wur.nl (T.V.); andre.bannink@wur.nl (A.B.)

**Keywords:** human edible protein sources, human inedible protein sources, breeding, protein efficiency, physiology

## Abstract

**Simple Summary:**

Livestock feed contains components that can also be consumed by humans, which may become less available for livestock. Proteins are such components that may become less available for livestock feed. This review focuses on using alternative protein sources in feed. We may expect protein efficiency problems and we discuss how these could be solved using a combination of alternative protein sources and animal breeding. We make recommendations for the use and optimization of novel protein sources.

**Abstract:**

Livestock feed encompasses both human edible and human inedible components. Human edible feed components may become less available for livestock. Especially for proteins, this calls for action. This review focuses on using alternative protein sources in feed and protein efficiency, the expected problems, and how these problems could be solved. Breeding for higher protein efficiency leading to less use of the protein sources may be one strategy. Replacing (part of) the human edible feed components with human inedible components may be another strategy, which could be combined with breeding for livestock that can efficiently digest novel protein feed sources. The potential use of novel protein sources is discussed. We discuss the present knowledge on novel protein sources, including the consequences for animal performance and production costs, and make recommendations for the use and optimization of novel protein sources (1) to improve our knowledge on the inclusion of human inedible protein into the diet of livestock, (2) because cooperation between animal breeders and nutritionists is needed to share knowledge and combine expertise, and (3) to investigate the effect of animal-specific digestibility of protein sources for selective breeding for each protein source and for precision feeding. Nutrigenetics and nutrigenomics will be important tools.

## 1. Introduction to the Review

The growing human population requires more sources for food production. The growth of income in developing countries will lead to an increase in consumption of animal-derived proteins [1]. Human edible protein sources (HEP) may become less available for livestock production in the near future. HEP can be defined as feed sources of plant origin that have a high enough appreciation and nutritive value to be directly consumed by humans, and humans and livestock compete for these resources. The livestock feeds are composed of HEP as well as human inedible protein components (HIP) such as by-products from the food industry. Proteins are important nutrients for cellular, tissue, and organ development, maintenance, and functioning, and if necessary, energy metabolism, for example in severe malnutrition or under extreme exercise conditions [2].

Two major strategies to handle the competition for protein sources with humans are (1) improving the efficiency of using feed protein sources and their conversion into animal protein (protein efficiency), and (2) replacing HEP with HIP. An important method to follow the first strategy is through animal breeding. In the second strategy, it is important to have knowledge of alternative protein sources, their potential to replace HEP with HIP sources, the consequences of using alternative protein sources on animal performance, and how to deal with these consequences. This global problem is recognized by the FAO [3], and also the Dutch ministry of Agriculture, Nature, and Food Quality published a policy note on the topic of a national protein strategy. It describes the diverse protein sources, the importance of protein sources, and the wish to change to locally produced protein sources [2].

The need to use alternatives for HEP also becomes prevalent due to societal demands. First, there is the need for a more circular economy, where no waste is produced, but instead all (end)-products are reused. Thus, both leftovers of human food and by-products of human food production should become available for livestock feed. Many of these products are already reused nowadays (e.g., rejected human food items), but there is much to improve for human food leftovers. Especially, the use of human food leftovers for animal feed production is forbidden by law because of the fear of animal disease outbreaks. While by-products of food production are often of lower feeding quality, and hence HIP sources, they may be available at low(er) cost. A second demand is to replace feed resources for which their global transportation comes with a major ecological footprint, favouring local feed products in the future [4].

The main objective of the present review was to investigate the potential use and optimization of HIP sources for livestock feeding. We explore the possibilities of breeding to improve protein efficiency using alternative feed sources that do not or to a lesser degree compete with HEP. We brought together knowledge on feed and protein efficiency of present-day livestock diets. This knowledge is combined with the genetic and biological mechanisms underlying protein efficiency, in particular in relation to changing livestock diet compositions. This review discusses the following topics: (1) the current situation for protein efficiency achieved by livestock, (2) the availability and value of alternative protein sources, (3) the physiology underlying protein efficiency, precision livestock farming, and nutrient partitioning, and (4) the future developments in livestock production. The review focusses on dairy cattle (*Bos taurus*), pigs (*Sus scrofa*), laying hens (*Gallus gallus*), and broilers (*Gallus gallus*). We use this knowledge to make recommendations to improve the public acceptability and sustainability of the livestock production chain.

## 2. Present Situation Protein Efficiency

The present situation is important as a reference point to evaluate the consequences of a transition in livestock protein feeding. Please note that data for the tables in this section were adapted from the CVB animal feed website [5]. In general, all species have HEP components in their diet. Due to the high percentage of grass products, and other roughage such as maize silage, dairy cattle diets have the lowest percentage of HEP. Poultry (layers and broilers) have the highest percentage [6]. On average, the diet of dairy cattle contains 15% crude protein (CP), of pigs 16–20% CP, and of poultry 16–23% CP. Animals of these three livestock species are fed different diets during the production cycle. Whole body growth and the development of specific organs require specific nutrients, and in particular amino acids (AA) from HEP or HIP are among the most important nutritional components. Soybean meal (SBM) and rapeseed meal (RSM) are among the most preferred protein sources for dairy cattle, pigs, and poultry. The preferred use of SBM and RSM may be country-specific and also may be because of the low cost of these components. Dietary protein concentration and nutrient digestion are important characteristics of HEP and HIP sources. Broilers are considered to have the highest protein efficiency and dairy cattle the lowest [6], which is partly due to the composition of the feed, with cattle having grass as a main constituent. Not only the use of HEP sources in the diet, but also protein efficiency of livestock determines the competition for the use of dietary protein sources with human food production. Replacing HEP with HIP components may affect the production level and efficiency of protein use. This may be due to the digestibility and bioavailability of feeds, and the efficiency of utilisation of protein resource as a feed. Van Krimpen et al. [7] calculated that replacing HEP with HIP increases the price of the feed in The Netherlands, and probably also in other countries, and decreases animal performance. The latter may be because the AA composition of different HIP protein sources is less optimal for livestock species because of their intrinsic amino acid composition. As a result, the animal metabolism may be affected. Some details are given below per species.

### 2.1. Dairy Cattle

Table 1 gives an overview of the presently used dietary sources for dairy cattle feed production. Stimulation of the intake of roughage and concentrates results in rapid development of the rumen, which is especially important for the development of the microbial population and rumen fermentation capacity. This results in increased uptake of feed, the development of rumination and salivation, and increased capacity of rumen epithelia and volatile fatty acids as end-products of microbial fermentation [8].

The two main steps in protein digestion by ruminants are (1) the microbial degradation of feed protein in the reticulorumen, and (2) the post-ruminal enzymatic digestion of feed and microbial protein in the small intestine. Microbial protein is the most important and reliable source of metabolizable protein (MP) for ruminants, because the AA composition is comparable to milk protein [9]. Dairy cattle dietary protein sources varies for the level of protein content and the ratios of rumen degradable (RDP) vs. rumen undegradable proteins (RUP). The diet contains protein sources in three groups: (1) grains, such as ground wheat and cracked corn (respectively high and low RDP), (2) intermediate protein, such as corn gluten feed and alfalfa meal (respectively high and low RDP), and (3) high protein, such as canola meal and fish meal (respectively high and low RDP). For details see [10].

The MP is a more reliable estimate of the aminogenic nutrient availability for the ruminant than CP because MP is the absorbed amount of protein that is net absorbed from the small intestine [9]. Feed CP contains two fractions of MP available for the cow: RDP and RUP. The RDP is used by the rumen microbes as an energy and nitrogen source to produce microbial proteins. The RUP fraction in the diet escapes microbial degradation in the rumen into the intestine. High producing dairy cattle utilize more AA, shaped as RUP, with increasing milk yield. However, clearly defined and additional AA requirements for dairy cattle during the lactating phase remains unknown. It is known, however, that methionine and lysine are the most limiting AA in high producing dairy cattle [11]. High producing dairy cattle may be supplemented with RUP sources or AA supplements resistant against rumen degradation in order to deliver more digestible protein to the intestine and increase MP allowance. Often, SBM is added to the diet in a rumen-protected form to serve as a source of rumen-bypass protein and increase the RUP fraction [12]. However, there are various alternative protein sources available to replace SBM as a protein-rich feed source. Dairy cattle diets contain a significant amount of roughage, fresh grass, silage grass, or hay, which contain HIP elements. In addition to roughage, there are different high-quality sources of both energy and protein feed of which maize silage and SBM are most common. A decreasing CP content of ruminant diets, such as with an increasing fraction of maize silage, is often accompanied with a higher allowance of SBM. The usual feed for dairy cattle consists of only 4.5% HEP. Decreasing this further reduces the amount of maize in the feed.

### 2.2. Pigs

Globally, commercial pig production systems are comparable; however, there are some differences in weaning age. Too low of a protein intake during lactation results in reduced milk yield and reduced fertility in later cycles. Weaned pigs are fed a starter diet up to 25 kg. Pigs receive a growing diet up to 55 kg and a finishing diet until slaughter weight at 115–120 kg [13]. Table 2 gives an overview of the presently used dietary sources for pig feed production. Over 50% of the pig feed is made of cereals and cereal by-products, which are high-quality starch sources. SBM, RSM, and sunflower meal (SFM) are high-quality protein sources. The diet of a pig consists of approximately 14–17% CP. The fraction HEP is approximately 25% [5].

### 2.3. Poultry—Laying Hens

During the rearing period of pullets, different diets are given to develop different parts of the body (Table 3). During weeks 14 and 15 pullets are fed ad libitum for growth and development of the reproductive tract. Pre-lay diets are fed until egg production has reached 2% in the flock. Feed consumption during the first few weeks of lay may increase by approximately 40%. Presenting the feed in crumb form instead of meal stimulates feed intake and BW gain. From three weeks of age, grit is provided to stimulate gizzard development.

The layer diets differ in the recommended protein and AA content depending on the average feed intake of the flock. Layer diet 1 includes higher AA content, which is required for production and BW gain. Additional dietary fat increases egg weight and egg weight reduces with reduction in dietary energy content, whereas it has little effect on the number of eggs produced. Feed consumption mainly depends on the energy requirement and the temperature. The AA requirement does not change throughout the laying period. Deficiency of one or more essential AA results in reduced performance. This can be accounted for by 2/3 of a reduction in rate of lay and 1/3 in a decrease in egg weight. Therefore, it is not possible to decrease egg weight towards the end of lay by reducing dietary AA concentration without a reduction in rate of lay. Table 4 shows the estimated requirements for standardized ileal digestible (SID) AA for laying hens to reach maximum production.

The daily AA requirement has been changed due to genetic progress over the last 30 years, which increased production by more than 40% while feed consumption decreased by 10%. There is an increasing trend towards formulation of low protein diets with supplemental free AA, which have become readily available commercially [14]. Currently, over 50% of the laying hen diet consists of cereals and fats as energy source, whereas SBM, RSM, and SFM are the main protein sources used.

### 2.4. Poultry—Broiler Chickens

Commercial broiler production consists of rearing of the parent stock, the laying period of dams, and the rearing of broilers. Each life phase has specific dietary requirements. The protein and AA contents in the diet and their ratio to energy content are important for parent performance, hatchability, and chick quality [15]. Broilers are grown to an average of 2.25 kg in approximately 35 days before they are slaughtered. Legume seeds are often used for supplementing protein. Due to the digestible AA composition, SBM is particularly important in broiler production [16].

Table 5 shows the estimated requirements for standardized ileal digestible (SID) AA for broilers fed ad libitum (in g/kg of feed). The average feed intake is 3.8 kg to produce a broiler of five weeks old with an average weight of 2.25 kg, and the average CP intake is 0.80 kg. In broiler chickens, the HEP components range between 21 and 24%.

## 3. Discussion

Table 6 summarizes the percentage of dietary HEP currently used in the diets of livestock, which is used as a reference point for evaluation of potential changes in feeding management. Animals of all livestock species are fed different diets during the production cycle. Whole body growth and the development of specific organs require specific nutrients, and proteins, in particular AA, are among the most important nutritional components. Dietary protein content and nutritional availability are important traits. Therefore, digestibility is an important trait for protein availability and protein efficiency, which may differ for different feed components. This may be related to the animal’s genotype and the gut microbiota, e.g., rumen digestibility of feed components.

All livestock species use HEP components in the diets. Due to the high percentage of grass and grass products, dairy cattle use the lowest percentage of HEP sources, while poultry (both layers and broilers) use the highest amount [6]. However, broilers are described as the most protein efficient species. This indicates that not only HEP sources in the diet, but also protein efficiency are important for the level of competition for HEP sources.

Replacing HEP with HIP components may affect production level and efficiency. This may be due to several reasons, including the digestibility of these feed components, and the protein efficiency of the feed. Next, we discuss the (experimental) knowledge of the HIP diets, and the possibilities for improving their use in livestock production.

## 4. Alternative Protein Sources: Replacing HEP with HIP Sources

Animal production can upgrade proteins to high-quality food for human consumption. Van Krimpen et al. [7] published a list of feed ingredients containing a wide range of protein sources as alternatives for soybean products and showed that the protein content of some novel dietary protein sources are lower than the SBM as the reference protein source (Table 7). It should be noted here that a considerable part of the protein supply in livestock comes from cereals, which are of course HEP but are not generally considered as such.

In searching for HIP as an alternative to HEP, the criteria for HIP are as follows: (1) the protein source should be able to perform well in the specific climate conditions of the specific country, (2) the cultivation of the protein source in Europe is currently not common practice, and (3) in the long term, the protein source is still applied in feed and not in food—i.e., it does not change from HIP to HEP [7]. We added that the production of the alternative feed source should not compete with human food production, e.g., for land area.

In most livestock the need to replace HEP sources with HIP sources is recognized. Feeding trials indicated different effects on productivity and mortality. The major classes of alternative protein sources include the following.

### 4.1. Insects

Insects can be reared on low-grade bio-waste, turning bio-waste into high-quality protein [17]. Production of insects does not require land, although the production of the feed of insects might, but needs attention from an energy use and global warming perspective [18]. Insects are part of the natural diet of poultry [19] and have a suitable protein composition [20]. Several insect species and their larvae are interesting as a feed ingredient for poultry diets due to their suitable protein composition [20]. The nutritive value for poultry and pigs is known for many insect species [21,22]. General data on AA composition and digestibility are promising, although the nutrient digestibility of (processed) insects and optimal inclusion levels as feed ingredients need to be further evaluated before implementation in practice. It is technically feasible to use insects as a sustainable, alternative protein-rich feed ingredient [21]. Legal barriers to feeding animal-derived feed components, including insects for livestock species, are presently limiting the use of insects in livestock diets. It should however be noted that insects are changing from HIP to HEP presently, since the human diet may contain more insect-derived protein in the near future, as willingness to pay for insect-derived products increases and processing improves [23,24].

### 4.2. Microalgae

Algae have the potential to be a highly productive protein source that can be produced efficiently. Marginal land (i.e., land of poor quality for agricultural production) is used for algae production, thus largely avoiding competition with agriculture [25]. The use of algae is currently limited to algae sources produced and processed in an open-pond system, which is vulnerable to contamination and has low productivity. Alternative systems have higher costs but may become feasible when used on a larger scale [26]. Widespread use is limited due to harvesting access and rights, seasonality and geographical locations, time consuming processes with production, harvest and transport, economic infeasibility in general and the potential of microalgae to gather toxic elements during production [27], and the potential negative effects on feed intake [28].

Approximately 30% of the world’s total production of microalgae is used for animal applications. Algae can be efficiently produced with nutrients from waste, e.g., manure, and solar energy [29]. The protein composition of algae varies among species [30].

Depending on the algae strain, microalgae contain 25 to 50% protein, although the real protein content in algae may vary between 6 and 12% if correctly measured [30]. Microalgae match conventional feed for CP content and AA composition except for the sulphur-containing AA, whereas lysine is more abundant.

Microalgae also contain non-protein-nitrogenous compounds, such as cell wall molecules, and oil and secondary metabolites, which may influence animal metabolism. Especially ruminant diets have the potential to become supplemented with distinct species of microalgae as a source of nitrogen [30], although feed trials have been done with other species as well. Widespread use is still limited due to numerous factors including harvesting access and rights, seasonality and geographical locations, time consuming processes, economic infeasibility in general, and the potential of microalgae to gather toxic elements while being produced [27]. In beef cattle, lipid extracted *Chlorella* species showed higher microbial efficiency than SBM in a diet containing 13% CP dry matter (DM) [31]. Inclusion of up to 20% lipid-extracted algae in the diet of wethers showed no change of growth performance and carcass characteristics compared with conventional diets [32]. Adding 1% to the diet of dairy cattle leads to a more mediated venous acid–base balance if acidosis occurs.

In pigs, no differences in FCR were observed when *Chlorella* and *Scenedesmus* replaced SBM in concentrations up to 10% of the diet of growing pigs. The BW gain of weaned piglets fed diets with Spirulina was higher compared to the control fed pigs, but this result was not consistent over all experiments [33].

In laying hens, Ekmay et al. [34] showed that feeding 26-week-old Shaver White laying hens with corn–soy–wheat isocaloric and isonitrogenous diets containing 25% defatted green (DG) *Desmodesmus* species biomass or 11.7% full-fat diatom *Staurosira* species (FD) biomass supplemented with or without protease for a period of 14 week did not affect egg production or body health. Ileal AA digestibility with DG- or FD-fed diets was higher than in those fed the control diet, and in the diets containing DG, it was higher than that of laying hens with diets containing FD.

Evans et al. [35] showed that 16% Spirulina algae replacing SBM resulted in an increased live weight gain between 3 and 21 days of age in broilers. Further increase of the algae content resulted in lower BW gain and total BW, possibly caused by the decreased feed intake of the broilers. Spirulina algae have a closer resemblance in AA composition to soybeans than corn, but there is a clear difference in the AA composition between Spirulina algae and soybean.

### 4.3. Seaweed

Seaweed or macroalgae contain up to 60% polysaccharides but also high value compounds such as colorants, omega-3 fatty acids, and bioactive compounds. Seaweeds contain 10–30% protein. Seaweeds suffer from some of the same production problems as microalgae. The essential AA composition of most seaweed species is not optimal for livestock nutrition, and all seaweed species show a high mineral content, which limits gross energy concentration on a DM basis. Macroalgae include Phaeophyceae (brown algae), Chlorophyta (green algae), and Rhodophyta (red algae). The nutritional value shows a broad range, depending on characteristics of the species. Brown macroalgae contain 5–13% CP on a DM basis and shows a very mineral-rich profile. Red algae contain 10–29% of CP, and green algae over 15% [36].

Feed trials with seaweed have been performed with dairy cattle and pigs. In dairy cattle, the essential (RUP) AA composition of most seaweed species is suboptimal [37]. Rumen degradability of CP in seaweed seems to be low and that of RUP high, which is generally considered beneficial in ruminant nutrition. This needs to be confirmed for seaweed. In growing–finishing pigs, there is a wide range of the DM digestibility of 26% to 71% for different seaweed species. In general, seaweeds are unpredictably variable as an energy or protein source for pigs [38]. Moreover, half of the pigs developed acute diarrhoea and refused to consume the diet further.

### 4.4. By-Products (Upgraded Food Crop Leftovers)

Leftovers from human edible crops can be used for livestock diet in all livestock species [39,40]. Table 8 summarizes plant by-products that have been used to replace SBM in the diets of livestock in several experiments. The table indicates maximum dietary content without affecting productivity. Depending upon the stage in the production cycle, replacing HEP with by-products shows species and age-related effects on performance and health of the animals. It should be noted that some of the feed sources in Table 8, such as field peas, have a high HEP content and are therefore not an optimal solution with respect to preventing competition with utilization of these sources as human food. *Camelina sativa* (CS) is an easy to culture crop, the seeds of which can be sprinkled on salads or mixed with water to produce an egg substitute and can also be used for the production of biofuels. By-products from CS are most favourable to ruminants. On average, the CP content is higher than in canola meal and contains a similar RUP content [41]. Guar is used especially for poultry and consists of 38–48% CP and 3–7% crude fibre. In dairy cattle, guar meal can be included up to 15% in the diet; however, 4% is optimal. Guar meal shows a rumen digestibility of 47% after 48 h, and thus guar meal may be a good alternative protein source in dairy cattle.

Vander Pol et al. [42] showed that 15% field peas could replace SBM and corn grain without changing milk yield and milk composition in lactating Holsteins. Milk N efficiency and concentration of milk urea was not affected. However, field peas have a high HEP content and are, therefore, not an optimal solution.

Froidmont and Bartiaux-Thill [43] replaced SBM with coarsely ground lupine and pea seeds in high producing dairy cow feed. Milk yield was lower with pea seeds, intermediate with a lupine and pea seed mixture, and higher with lupine and SBM diets, and milk fat increased with lupine. If SBM was completely substituted by lupine, similar milk yields were observed but milk fat was reduced, probably caused by the lipid content of lupine. Nitrogen efficiency was not affected. They concluded that lupine seeds could replace 75% of SBM on a DM basis. Goncalves [44] showed that SBM can be replaced with different sources of urea as a nitrogen source without changing the productive performance and milk composition. DM intake, neutral detergent fibre, organic matter, CP, and total digestible nutrients were not affected. The average digestibility was 65%. They concluded that substitution of SBM by 2.1% coated urea on a DM basis is possible. In recent years, accessibility to canola meal, which is derived from RSM, as a protein source for dairy cattle rapidly increases. Broderick et al. [45] showed that replacing SBM with canola meal increased feed intake, milk yield, and true protein for either low and high CP content, and rumen-protected methionine and lysine intake increased without affecting production. Drackley and Schingoethe [46] showed that SFM replacing SBM results in a higher milk protein percentage, probably due to a more desirable AA balance in diets. SFM has approximately the same amount of CP content and an even higher RDP percentage compared to SBM. In pigs, Landero et al. [47] showed that up to 200 g rapeseed expeller/kg can replace SBM in diets starting 1 week after weaning without reducing growth performance. However, increasing inclusion of rapeseed expeller linearly reduced the apparent total tract digestibility of energy, DM, and CP, and the digestible energy content of diets. Landero et al. [48] substituted SBM with solvent extracted *Brassica juncea* RSM in nursery diets starting 1 week after weaning, and these pigs were lighter than pigs not fed juncea RSM. *Brassica juncea* RSM contains more glucosinolate and gluconapin and a lower fibre content than the conventional rapeseed. Nørgaard et al. [49] evaluated SFM, rapeseed cake, and field pea replacing SBM in pigs over 35 kg and reported that the SID of CP and AA was lower in SFM and rapeseed cake than for pea. They concluded that SFM, rapeseed cake, and pea can be used in diets for pigs as alternatives to SBM. González-Vega and Stein [50] showed that the SID of most AA in rapeseed, cotton, and sunflower products were less than in SBM in pigs over 107 kg. Hanczakowska and Swiatkiewicz [51] concluded that a mixture of rapeseed press cake/legume seeds can replace SBM in fattener diets without affecting growth performance. Okrouhlá et al. [52] concluded that 12% RSM replaced SBM without affecting growth or carcass characteristics and meat quality. McDonnell et al. [53] concluded that RSM can be used up to 21% in the diet as a direct replacement for SBM with no associated depression in performance, when formulated on an ileal digestible AA and NE basis in pigs over 40 kg. Shelton et al. [54] showed that average daily gain (ADG) and gain:feed ratio of barrows and gilts fed diets with 49% RSM was lower than pigs fed 32% SBM during the growing phase.

Shi et al. [55] showed that replacing SBM with SFM did not affect performance and egg quality of laying hens; however, after six weeks the egg yolk cholesterol concentration was lower in birds fed SFM. Young birds are sensitive to high-fibre content in their diets. The crude fibre content of SFM may be up to three-fold higher than in SBM, and because the fibre is highly lignified, it is resistant to bacterial degradation in the digestive tract. Air classification differentiates the protein (fine fraction) and fibre (coarse fraction) particles. The SFM low-fibre fraction may have better feeding value for monogastrics, while the high-fibre fraction could be intended for feeding ruminants. Laudadio et al. [56] used air-classified SFM replacing SBM, showing unaffected growth performance, improved feed consumption, efficiency, and egg production, including a higher percentage of medium-size eggs. It was concluded that SFM reduced the production costs.

Rutkowski et al. [57] fed narrow-leaved and yellow lupine (*Lupinus angustifolius*) pea diets containing 16.0% CP and 11.3 MJ ME/kg, replacing SBM as a protein source and showing that 27.5% of legume seed in laying hen diet affected performance results negatively, but 19.5% of these seeds and 8% RSM in diets could be accepted as an SBM substitute. Van Krimpen et al. [7] showed that lupines can be used in laying hen diets up to an inclusion level of 20% without affecting production performance. Koivunen et al. [58] showed that feeding laying hens with white-flowered semi-leafless green spring peas did not affect production performance and egg quality and concluded that at least 300 g/kg peas can be used in diets. They also showed that replacing SBM with unprocessed and expander-processed faba beans did not affect egg production rate, egg mass production, feed consumption, or FCR, but it decreased egg weight and egg exterior quality, and it tended to increase hen mortality. Faba bean processing had no effect on egg production parameters or hen mortality. The authors concluded that 50 g/kg faba beans can be used in the diets of laying hens without negative effects on production performance or liveability. It is recommended to use tannin-free varieties in diets for monogastrics and cultivars with low levels of vicine/convicine in poultry diets [7].

Leiber et al. [59] partially replaced SBM with a mix of three substitutes (alfalfa, peas, or meal of the black soldier fly larvae), showing a similar performance for average daily gain for all diets in broilers. Dotas et al. [60] showed that field peas replacing SBM did not affected performance. However, pulse grains such as field peas are highly digestible for humans and are therefore not an optimal substitute for SBM. In young broilers, field peas and faba beans gave better growth rate and feed efficiency. FCR improved with increasing amounts of faba beans in the diet. However, broilers older than 21 days showed negative growth rate effects [61].

### 4.5. Animal Protein Sources

Animal protein sources are attractive due to a more balanced protein composition in comparison to plant-based protein sources. Limited availability, high price, and legislation in various western countries impact its use. Nevertheless, for example the use of fish-meal in broiler feed does increase the growth rate and feed efficiency in broilers as compared to vegetable protein-based diets [62]. Therefore, fish discharge is commonly used as animal feed [63,64].

### 4.6. Food Wastes

Human food wastes can be fed to animals (e.g., see for example the Kipster farm, which has been described as the most animal, human, and environmentally friendly chicken production system in the world [65], and related initiatives for pigs). This may also be a source of pathogen spread, which limits application with regard to controlling food and feed safety. Therefore, this will not be discussed further.

### 4.7. Discussion

All HIP sources used have limitations compared to HEP sources. Therefore, at the moment it will not be possible to make the feed of livestock completely non-competitive with human food. A model system including all variables is highly needed to guide and optimize feeding trials and to enable one to compare and predict the outcome of mixed alternative protein sources in the animals.

It should be recognized that several of the mentioned HIP sources do compete with the production of HEP sources, e.g., for use of crop land. Although these protein sources themselves are indeed human inedible, they are still affecting human food production. For this criteria, insects, algae, and seaweed are the least competitive protein sources, although seaweed and insects also have been mentioned as HEP sources.

It is expected that there will be variation among animals how they can cope with different sources in the diet. Furthermore, processing of the protein sources might affect the protein efficiency of livestock production by completely replacing HEP by HIP without reducing livestock performance and cost-effectiveness.

Conclusion: For dairy cows, SBM remains one of the most used and preferable sources of protein. It has a very high CP content and a well-balanced AA composition, and it also has a relatively large proportion of RDG [66], but possibilities for replacements have been shown. RSM can be included in diets for pigs to a certain extent to replace SBM, but the glucosinolate and gluconapin of the rapeseed source should be taken into account. For chicken, not all HIP sources are realistic due to specific AA requirements.

In most important livestock species, the need to replace HEP sources with HIP sources is recognized. Independent feeding trials indicate different levels of effects on production and mortality of the animals. Unfortunately, there is no standardized experimental design, making it difficult to compare the different feeding trials. Therefore, a standardized experimental design (per livestock species) needs to be developed.

## 5. Preparing for Livestock Feed Trials

To determine the consequences of decreasing HEP in livestock, a general feed for each livestock species has to be determined. A diet is required to fulfil the physiological needs of the animal. The experimental diets should be isocaloric and contain the required levels of digestible AA. The Bestmix linear-programming model determines the composition of the feed by the cost price of feed ingredients [67]. Therefore, the model has been set at minimizing the ingredient costs. An estimation of the percentage human edible content per feed ingredient was made in 2007. Table 9 shows the HEP content of livestock diets. The amount of HIP can be calculated by multiplying the percentage human inedible with the CP content of the feed ingredient.

The standard starter feed for pigs [5] contains about 25% HEP, and therefore 75% of the feed is human inedible. Formulating diets to include 80–90% HIP based on least cost formulation resulted in a more than double cost price of the feed, based on Dutch diets in 2013. The standard feed for laying hens is 70% HIP. When HIP was increased to 90%, the price of the feed increased more than 60%. The HIP content of broiler feed ranges from 76 to 79%. A rather small increase in HIP content might dramatically increase the feed prices already, because an unrealistic high level of free AA is supplemented to fulfil the requirements. Ingredients with lower levels of protein can be used, thereby giving more flexibility to select protein sources with a higher HIP content. The dairy cow feed contains only 4.5% HEP, and therefore 95.5% of the feed is human inedible. Further increase of the percentage HIP toward 98% decreases the percentage of maize and the cost price of the feed increases further by approximately 50%.

Two other aspects of using HIP require further investigation: (1) sensory aspects of HIP-containing feed for the appreciation of the livestock related to willingness to eat and appetite, and (2) sensory aspects of the animal-derived human food products due to the use of HIP in the animal feed. Presently, knowledge of this is lacking, and positive associations between livestock appetite for HIP-containing feed and human sensory evaluation of the derived food products could stimulate the use of HIP in livestock feed.

## 6. Protein Physiology of New HIP Sources—Relation with Protein Efficiency

Metabolizable AAs become available from feed or from microbial proteins after digestion in the stomach and the gut by proteolytic enzymes provided by the animal and the gut microbiome. Livestock species differ for proteolytic mechanisms. In cattle, rumen protein digestibility is an important characteristic to estimate the availability of feed N for microbial N synthesis [68]. Both rumen undigested feed protein and rumen synthesized microbial protein contribute to the ileal digested protein. Both contribute to the part that remains undigested in the small intestine and contributes to excreted N and N emissions to the environment. Highly efficient protein digestion and absorption from the feed is important for a cost-effective production and for societal acceptance of livestock. Changes of dietary sources (i.e., from HEP to HIP) may have important consequences for rumen protein degradability and microbial protein synthesis, and intestinal digestibility of the rumen bypass fractions of feed protein [69,70]. There may be important differences between individual animals. The proteolytic mechanisms are less complex for monogastric animals. Change in dietary protein source may also affect voluntary feed intake, appetite, and required feed volume.

Protein efficiency is currently not presented as a separate trait but may be derived indirectly from observations on feed efficiency. We must mention that of course protein efficiency is not the same as feed efficiency, although a good correlation may be possible. So far, there is no trait that can be used as an indication of protein efficiency. However, feed efficiency is the closest related trait most generally measured. Using proxies such as (labelled) AA metabolism may be (at the research level) an alternative. These are not considered here because we focus on the practical farm level. Koch et al. [71] proposed to adjust feed consumption of growing animals for BW gain and mid-weight (residual feed intake (RFI)) to evaluate feed consumption of individual animals with energy requirement for gain and maintenance expected to be equal [72]. Various other traits are used, however. Table 10 gives an overview of the reported heritability of productivity, digestibility, feed efficiency, RFI, and other traits of the livestock species, which is the basis for the discussion per species below.

### 6.1. Dairy Cattle

Feed efficiency (FE), expressed as kg milk produced per kg of dry matter intake, is an important production trait. In cattle, FE should be high for the whole lactation period. If FE is very high (>1.7 kg of 3.5% fat corrected milk divided by kg of DMI) [89,90] at the start of the lactation, the energy deficit may be excessive, related to a low level of feed intake compared to milk yield, which is driven by the mobilization of body reserves. Because individual feed intake is mainly collected in research herds or nucleus breeding herds, large-scale data on individual feed intake is limited [89,91]. Differences in RFI among individual cows within herds exist [92].

As an alternative to protein efficiency derived from FE, milk urea nitrogen (MUN) reflects the efficiency of N utilization and the N output towards the environment [87]. Milk urea originates from an imbalance between dietary N and energy for microbial growth in the rumen and inefficiency of the utilization of absorbed protein from the intestine [93]. Urea in milk, blood, and urine is related to the amount of CP and energy in the diet [94]. Genetic selection and adjustments in management both can influence MUN [87]. Selecting for low N sires will result in progeny with lower milk MUN levels compared to the daughters of the average bull. It remains to be established whether MUN heritability is causally related to protein efficiency.

### 6.2. Pigs

Body composition traits are a proxy for leanness, which can be used to improve FE of growing animals. The $h^{2}$ of traits related to FE of pigs is shown in Table 10. More than one third of the variation in feed intake is due to body maintenance processes, including basal metabolism, protein turnover, thermoregulation, physical activity, immune and other coping functions, nutrient digestion, and absorption efficiency [95]. Models describing one of the biological processes involved in nutrient utilization or energy needs have been developed to improve FE. Total feed efficiency (TFE) models could be developed describing the total system to produce a finisher pig, including the total feed intake of the sows, piglets, and finisher pigs and the total output in kg. The most important traits correlated with improving TFE are back fat thickness, ADG, litter size, litter mortality during lactation, and BW of the sow at the start of lactation [75].

Traits most often included in the breeding objective for selection of efficient lean meat growth are growth rate, back fat thickness, and FCR or feed intake. Selection for high leanness and a low FCR under ad libitum feeding leads to a reduction in feed intake in the long term, limiting the potential to deposit lean meat growth. Therefore, efficiency of protein deposition can only be measured in conditions where AA intake just meets the AA requirements of the growing pig, enabling one to determine the genetic potential for maximal protein efficiency. This indicates there is a strong dependence between nutritional conditions met and expression of the genetic potential by animals.

### 6.3. Poultry—Layers

Feed intake, FCR, and RFI are important breeding traits in laying hen breeding. Genetic improvement for egg production in commercial layers is approximately 2.5 eggs per year [96]. Ten to thirty percent of the variance of daily feed intake (DFI) remained unaccounted for by metabolic BW, daily egg mass production, and BW gain (i.e., RFI) of hens showing equal production levels, whereas BW differed considerably with regard to feed consumption and FE. This may suggest metabolic losses.

### 6.4. Poultry—Broilers

Breeding for FCR resulted in a reduction in feed requirements of approximately 0.013% per year [97,98]. FCR is often measured on the farm for management purposes, and it is on average 1.68. Selection of FCR is profitable in broilers due to its relatively high heritability [82]. Ileal AA digestibility (protein digestibility) and protein deposition in the body are determined to measure protein efficiency. Mignon-Grasteau et al. [99] measured protein digestibility in the manure, which includes urine. The manure contains a mixture of endogenous proteins (e.g., excreted enzymes), epithelial cells, bacterial protein, and the residuum of digested feed [100]. Therefore, manure is also not a direct reflection of efficiency of feed protein digestion as an important factor affecting protein efficiency.

## 7. Genotype × Nutrition (G × N)

Animal performance (e.g., growth rate, milk yield, egg yield) depends on the interaction between genotype and environment (e.g., feeding level, feed composition, housing conditions) [101]. Several studies, focusing on the different livestock species, found G×N interactions [101,102]. Selection improved animal protein efficiency considerably. However, this was done using high-quality feeds, and the best performing animals with these high-quality feeds are not necessarily the same animals as those best performing on the alternative protein sources. Therefore, it is important to consider the relation between the animals’ genotypes and the diet provided [103]. The environment can also influence the expression of the genotype via epigenetic modifications such as DNA methylation and histone modifications to induce mitotically heritable changes in gene expression without altering the DNA sequence [104,105]. In gene promoters, methylation generally leads to decreased levels of transcription via alteration of transcription factor binding or changed chromatin conformation [106,107], while gene body methylation is generally associated with increased levels of transcription [108,109,110]. The methylation level of genes and genomes has been shown to be significantly associated with complex and disease traits [111,112].

### 7.1. Dairy Cattle

Larsen et al. [113] studied lactating Holstein Friesian (HF) and Jersey cows fed a diet supplemented with a linseed and rapeseed mixture. Milk production, fat %, and lactose % were affected in Jerseys only, suggesting a G × N interaction. White et al. [114] showed an interaction between breed and environment for milk production and milk protein % in Jersey and HF cows consuming two different diets. Similarly, Roche et al. [115] indicated the importance of the genotype × environment interaction. These studies suggest that breed should be considered when applying nutritional changes, and vice versa, and when selecting for genotype, the nutritional conditions met are preferably taken into account. It is recommended to select animals adapted to various feeding conditions in order to improve protein efficiency.

### 7.2. Pigs

Animal growth performance and pork quality depend on the interaction between genotype and environment (e.g., feeding level, composition, housing conditions). Growth performance of different pig breeds can be different with varying feeding levels. Affentranger et al. [116] found that the feeding regime mainly determined feed intake and daily gain, and that maintenance requirements differed between different genotypes. Wood et al. [117] found effects of breed and diet on growth rate and carcass composition. Fuller et al. [118] showed that pigs from different breeds or genetic populations either fed a low or high protein diet may differ in their lean growth potential or in their marginal response to nutrient intakes. On the contrary, in other studies no G × N interaction was found comparing different pig breeds fed either a high protein or a low protein diet [119], although lean-type crossbred pigs grew more efficiently on a high protein diet than a low protein diet. Pigs held in an environment with high temperatures have a lower feed intake (AA intake) than pigs in an environment with moderate or low temperatures. So, it is important to select pigs for maximal protein efficiency under different environmental conditions.

### 7.3. Poultry—Laying Hens

Strains differ in FE due to genetic differences in physical condition, basal metabolic rate, body temperature, body composition, and physical activity [120]. Singh et al. [121] found interactions between environments, strains, and ages on egg production, BW, and egg quality, suggesting that strain should be considered when applying different nutritional changes. Toprina G (yeast) inclusion in diets for two hybrid strains reduced FCR for one of the hybrid strains while FCE was unaffected in the other hybrid strain [122]. Interaction effects between dietary CP content and strain were observed for BW, yolk and shell percentages, haugh units, and albumen height. The BW, haugh units, and albumen height were more responsive to dietary CP content in hens [123]. On the contrary, Pérez-Bonilla et al. [124] found no effect of CP content and initial BW on egg quality traits in brown egg-laying hens. Santos-Ricalde et al. [125] reported that CP content affected feed intake and FCR but not egg quality traits.

### 7.4. Poultry—Broilers

The FCR differs within broiler strains and is likely to differ more between separate strains [64,65]. Reyer et al. [126] observed differences in FCR by quantitative trait locus analysis within one broiler strain. Part of the differences between strains could be attributed to gastric functions because the proventriculus and gizzard were bigger in strains with higher digestion values. Strains differ in production [127], protein and AA requirements [128], mineral requirements [129], and in digestibility of digesting different strains of wheat [130]. The FCR of the commercial broilers was lower than the hybrids, mainly due to the lower feed intake and higher daily gain [131].

## 8. Concluding Genotype × Nutrition

Selective breeding focusing on G×N interactions might be applied to improve protein efficiency. Recent projects like “Feed-a-gene” [132] tested the consequences of different diets on the results of selection programmes to select animals with a higher efficiency in general or under specific feeding conditions [133,134,135,136,137]. It is recommended to select animals adapted to various feeding conditions in order to gain protein efficiency.

## 9. General Discussion

The literature reviewed above showed us that the complex interactions make protein efficiency a difficult trait to unravel. Factors influencing protein efficiency might be (1) precision feeding and nutrient partitioning, and (2) nutrigenetics and nutrigenomics in the context of environmental–feed interactions and the amount of (voluntary) feed intake of alternative protein sources. As indicated, changing the source of protein in the feed of livestock may affect the protein digestibility, and therefore the protein availability for the animal and the excretion of nitrogenous compounds to the environment. Different protein sources influence performance, namely FE, FCR, and RFI, because of the direct relation between ileal protein digestibility and protein deposition. We discuss the differences between livestock species, heritability of protein efficiency traits, the importance of the interaction between the animal’s genotype and the protein source, and the composition of the gut microbiome as determining factors.

### 9.1. Protein Efficiency in Relation to Precision Feeding and Nutrient Partitioning

Protein efficiency is the net result of protein digestibility and net absorption of AA from the intestine and AA metabolism in the animal. Comparing ruminants and monogastric animals shows that individual essential AA is of particular importance for monogastric animals receiving low CP diets, since microbial protein is not a major source of AA, like it is in ruminant species. Alternative protein sources may differ in AA composition and protein digestibility. As a consequence, the absorbed AA profile may differ, leading to metabolic changes and a different AA partitioning, affecting livestock productivity and health. These relate to both breed-specific and individual-specific traits. Therefore, specific feed formulations including AA composition, percentage, and digestibility are required for breeds and for individual animals within breeds, which may differ from each other in terms of age, BW, and production potential [126,138]. While the first is common practice already, the latter is more difficult and is referred to with the term “precision feeding”. Animals need to be supported by different nutrient formulations to achieve the objectives with more precision and without loss of production potential, minimizing the loss of nutrients to the environment. Zhang et al. [139] determined that providing a diet to a pig adjusted to its BW lowers average feed costs because less protein is wasted. Implementation of precision feeding requires (1) evaluation of the nutritional potential of feed ingredients; (2) precise determination of individual animal nutrient requirements; (3) formulation of different premixes; and (4) adjustment of the dietary supply and concentration of nutrients to match the requirements of each individual in the herd [138]. The fulfilment of the fourth requirement is difficult and expensive in poultry, as they are kept in large groups hampering the practical feasibility of precision-feeding. Moreover, broilers lives are short to gather the required information for precision livestock farming and to adapt in time.

Digestibility is affected by changes in the passage rate affecting the time available for digestion and absorption in poultry [140]. Feed passage rate and digesta volume are animal-related factors affecting maximum daily feed intake. Digestibility is a major source of variation in FCR in cattle [141], often displayed as the digestion coefficient affected by intake, digesta retention time, age, and feed processing [142]. When the level of feed intake relative to maintenance increases, the digestion of feed (as measured by total tract disappearance) tends to decrease in ruminants. Feeding levels can affect both apparent and true total tract in piglets [143]. These authors investigated ileal and total tract digestibility in relation to feed intake levels and found that estimates for true digestibility were lower compared to their apparent values. They concluded that their results suggested that feed intake level can affect both apparent and true total tract nutrient digestibility in piglets.

Next to the decreasing digestion with increasing feed intake in ruminants, there is also genetic variation in total tract digestion of feed. On the contrary, studies in genetic differences in digestibility of monogastrics indicate that differences in digestibility are not important sources of variation in RFI [144], indicating species-specific differences between ruminants and monogastrics. However, it should be noted that there are indeed genetic differences underlying digestibility. Differences in digestive mechanisms are suggested as the biological background of these species’ differences. Different rumen and gut microbiomes may be important factors too. There is an urgent need to establish the relationship between rumen and gut microbiomes with digestion realized by animals.

It is particularly relevant to consider nutrient supply for the various physiological functions that need to be maintained. Nutrient partitioning significantly affects metabolism in production tissues such as muscle, reproductive organs, and the mammary gland [141]. It has been suggested that only 19% of the variation in RFI among animals is attributable to differences in diet digestion, with the remainder purported to be due to differences in physical activity, body composition, protein turnover and metabolism, and metabolic processes related to individual-specific maintenance requirements and nutrient utilisation for growth, milk yield, or other productive functions [144]. The biological effects of precision feeding for productivity may be limited by a changed nutrient partitioning when animals are challenged by the environment. For example, heat stress will decrease feed intake, resulting in a changed nutrient partitioning, and diseased animals produce high levels of several acute phase proteins [145] utilizing a large proportion of available AA [146]. To overcome such environmental challenges, it is suggested that more robust animals have to be selected. Next, protein efficiency may be improved for the robust resilient animals that easily adapt to different environmental conditions. New traits need to be developed to breed for more robust animals, particularly when the aim is to use alternative HIP sources, taking into account the AA availability from these sources.

By selecting animals in a specific environment, nutrients will have been reallocated between the important traits to produce and survive in that particular environment [147]. This implies that other traits may be adversely affected due to reallocation of nutrients, which is proportional to the heritability of the “resource allocation factor” defined by the proportion of resources devoted to production vs. maintenance [84,148]. Adaptation to an environmental challenge requires metabolic changes. During an infection, the priority of nutrient partitioning redirects nutrients away from production tissues to support the immune response [149]. Humphrey and Klasing [150] observed in chickens an increase in the requirement of lysine for the immune response. In great tits (*Parus major*), a 9% increased basal metabolic rate and 3% weight loss after one week following activation of the immune system with a non-pathogenic challenge was observed [151]. An intricate interface between the immune system and metabolism was found in more species, including chickens [152,153]. Negative correlations have been observed between production and fitness-related traits, such as fertility and health [154]. Such findings indicate that AA can be used only once, and the environment may direct repartitioning of AA utilization as a leading regulatory mechanism. In lactating animals, poor body condition scores during the period of negative energy balance after calving resulted in decreased fertility, probably due to metabolic stress [148]. The effect of negative energy balance may be partly compensated for by improving the environment, including an increase of dietary protein. Animals are adapted to the environment they are selected in, probably caused by a G×E interaction. Kolmodin et al. [155] showed in Scandinavian dairy cattle that animals with the highest observed production are the animals with increased sensitivity to the environment.

What would the influence of replacing dietary HEP with HIP or increasing dietary protein due to the lower protein quality of HIP be? Selection for higher production may affect the adaptation capacity to unexpected environmental changes. Similar breeding programmes at different geographical locations result in animals with different nutrient partitioning [147]. In a meta-analysis of selection experiments, Van Der Most et al. [156] determined that selection to increase growth rates in poultry lines decreases the immune function, however, with an unchanged cellular and humoral immunity. In contrast, selection for an increased immune function did not appear to have affected growth. This is in contrast with the above reported interaction between nutrition, general metabolism rate, and growth rate [152,153]. This suggests that with the correct nutrition, the negative interaction can be broken. Balancing breeding goals with regard to the nutrient partitioning seems therefore a logical option for improving efficiency, as has been demonstrated in beef cattle [157].

In conclusion, feed composition, precision feeding, and nutrient partitioning affect the protein digestion and the animal’s metabolism and performance. Dietary inclusion of alternative HIP protein sources may hence affect the animal’s metabolism through changes in metabolic rate and activity, thereby affecting economic traits including health and productivity. Furthermore, gut functionality is regulated by the animal’s genotype and microbiome, which are important determinants. Nutrient partitioning as a major source also relates to the environment, as simultaneous with the impact on the animal is also the impact to the environment changes due to changed excreted urine and faeces. Thus, it is concluded that optimized selection of animals and precision feeding of animals have positive effects on the environment as well (higher efficiency, less emissions), potentially improving livestock’s social acceptability. Replacing HEP by HIP will add to this social acceptability. This makes this objective much wider than just the cost-effectiveness of livestock operations. However, the sizes of the effects are presently unknown and demands further research in this area.

### 9.2. Genotype × Nutrition (G × N): Nutrigenetics and Nutrigenomics

Breeding can improve the protein digestibility of alternative HIPs. Separating the breed-specific and animal-specific effects show the underlying genetic component of protein efficiency. Therefore, each breed of interest needs to be tested independently in feed studies that aim to evaluate consequences of replacing HEP with HIP. Individual animal genetic variation is underlying the biological mechanisms important for precision feeding. Because some animals better deal with low quality protein sources than others, it is important to consider G × N interactions. The nutrigenetic G × N for protein efficiency is poorly investigated. Nutrigenetics mainly focuses on the effect of genetic variation on the interaction between nutrient requirements and traits including diseases. Genetic variation may also be important for differences in nutrient requirements [158,159,160,161]. Hence, species-specific as well as breed-specific nutrigenetics research is needed.

Nutrigenomics provides a molecular understanding of how nutrients affect animal growth and health by altering the gene expression of an individual’s genetic make-up [160]. Nutrigenomics focuses on the effect of nutrients on the regulation of the (epi)genome, transcriptome, proteome, and metabolome to highlight the metabolic/nutrient status of an animal and to provide knowledge on the nutrient use of individual animals to optimize protein efficiency. However, research on the application of nutrigenomics to demonstrate an improved performance and FE in livestock is still limited. Micro- and macronutrients can have important regulatory effects on gene and protein expression and metabolism. Small changes in the molecular structure of a nutrient can have a profound influence activating nutrient sensor pathways, which explains why closely related nutrients can have different effects on cellular function [160]. Further research on the consequences of replacing HEP by HIP sources will allow one to make more and better use of HIP as a replacement of HEP sources. In conclusion, research on tailor-made nutrition and diet-specific nutrigenomics and on animal selection (nutrigenetics) for each HIP source will lead to knowledge about the biological mechanisms underlying the individual animal-, species-, and breed-specific responses and economic possibilities to increase efficiency of livestock production in the future in a sustainable way.

## 10. Future Livestock Production: Potential Directions to Go Forward

The availability of HEP sources may become limited for livestock production in the (near) future. Improving protein efficiency of HIP sources can limit the need for HEP sources. The HIP sources may be introduced, although presently they can only partially replace the HEP sources. It should be taken into account whether and how these HIP sources require (less) land use. A combination of actions may be required for a transition in the use of protein resources. Precision feeding will reduce the amount of feed resources wasted. However, especially at the individual animal level, much knowledge is still lacking. Genomic information about the specific feed requirements of animals may push precision feeding forward. Proxies, or indirect traits, may be helpful to monitor the effects of changing diets on metabolism and economic traits, including health and productivity. While some proxies such as (labelled) AAs in the feed may be useful only in a research setting, proxies for use at the farm level should also be developed. Dedicated studies in the lab investigating the metabolism of livestock during the transition of alternative protein sources should come up with such proxies. Such information may vary among alternative feed components and for combinations of feed components and also relates to the environment for an optimal performance of the animals. Animals may adapt to the alternative feed, which requires more longer-term studies to be performed. Breeding for improved performance requires supported by nutrigenomic data, enabling a diet-specific genome selection within a specific environment. In the broiler industry, animals live too briefly to gather the required information for precision livestock farming. Therefore, breeding to reduce the animal variability at this point may be a better option. For longer lived animals, genomic information about the specific feed requirements of animals may push precision feeding forward. This may vary among all (alternative) feed components and combinations of feed components and relate to the environment and optimal performance of the animals. Furthermore, experimental duration needs to be studied, as animals may adapt to the alternative feed. Metabolic stress and environmental sensitivity will be highlighted in the nutrigenomic data, which hence may help to optimise animal–diet–production–environment interactions.

## 11. Conclusions and Recommendations

It is important to consider alternative protein sources and improve protein efficiency of livestock. Presently, increasing HIP in a livestock diet substantially increases the cost price of the diet [7], but this may change in the (near) future. The cost price may change due to three reasons: (1) Competition with humans for HEP may increase the price of HEP relative to HIP; (2) the circular economy may cause other products of the human food chain and other alternative protein sources to become cheaper; and (3) the required reduction of the ecological footprint of livestock reduction may lower the price of locally produced protein sources relative to higher quality protein sources imported from far way countries. Using less expensive and alternative sources like insects, seaweed, algae, locally grown sources to reduce transport costs, and upgraded feed crops and by-products will affect the price of the diet. A substantial amount of knowledge for the practical use of alternative protein sources is currently lacking, although some experiments have given promising results. These experiments have major experimental design differences, which hamper their direct comparison. The first recommendation is hence to develop a standardized experimental design including animal numbers and isonitrogenous and isoenergetic diet formulations, and to study animal-specific effects with nutrigenomic and performance measurements including feed digestibility and efficiency measures. Note that the optimal AA profile of an animal might differ with age and productivity level. Decision support tools with sophisticated models and a strong empirical basis could be used to evaluate impacts of alternative HIP sources (with HEP as a reference) on protein and energy supply, and protein and lipid deposition in livestock. A decision support tool could be used to investigate the impact of alternative sources on protein and energy supply, and protein and lipid deposition in livestock [162]. Such a tool is also helpful to recognise interactions; for example, van Milgen et al. [162] showed that protein deposition and lipid deposition depends on energy supply, even though the AA supply is sufficient. These models have been adapted to include a representation of G × N in case this information is still lacking.

Presently, there is still a large gap in approach between animal breeders and nutritionists. The second recommendation is hence to start more intensive discussions between animal breeders and nutritionists to share knowledge and combine expertise and start more joint experimental research. It is expected that this will lead to combined improvements in the formulation and manufacturing of feed ingredients dedicated to the individual animal to start precision feeding to better utilise proteins, reduce emissions, and improve livestock production sustainability from a societal viewpoint. Welfare and feed efficiency/nutrient partitioning goals are often seen as opposing goals. Nevertheless, welfare and protein efficiency might not have to be opposite goals if the environment is taken into consideration, with each environment having its own requirements. At the moment, such research approaches are scarce, and knowledge is lacking.

Selective breeding is a powerful tool towards a more sustainable livestock production. However, so far, no traits have been defined that could be used directly in a breeding program to improve protein efficiency. Indirect factors can be helpful here. The most important factors affecting protein efficiency can be divided into input-related (nutrition), animal-related, and output-related (production, excretion). Input-related factors are AA profile of nutrition, CP in the diet, HEP or HIP protein sources, feed intake, number of meals, energy/protein ratio, and supplementation with synthetic AA; animal-related factors are digestibility, gut health, protein turnover, nutrient partitioning, gut microbiome, age, variation, and dynamics within a herd or flock. Output-related factors are kg of milk, egg numbers, kg of meat, offspring, urine, faeces, and enteric emissions. The third recommendation is to investigate the effect of animal-specific digestibility, nutrient partitioning, protein turnover, and within herd or flock animal variability on protein efficiency, with an emphasis on the use of HIP as replacement for HEP sources. Nutrigenomic data recognise the animal-specific effects of different feed formulations. Minimized nutrient supply forces the animal to show its maximal potential that is genetically determined. For broilers, selection on uniformity to reduce the individual animal variation in protein efficiency seems the best option to improve the overall protein efficiency of a farm in practice. Finally, as a fourth recommendation, it is important to start to investigate the genetic correlation between protein efficiency and energy efficiency, especially when using HIP sources. Only if the genetic correlation is high is there less or no need to select for protein efficiency specifically. It is important to investigate overall protein efficiency, taking into account the protein efficiency during the entire life cycle of livestock.

## Figures and Tables

**Table 1 animals-11-02293-t001:** Main sources used in dairy cattle diets and the average nutrient composition.

		Nutrients (%)			
Resource	Energy Value (VEM) ^1,2^	Crude Protein CP ^3^	Crude Fat CF ^4^	Crude Fibre CFi ^5^	Starch + Sugars ^6^
Grass, fresh	1006	22.7	4.4	22.8	9.6
Grass, silage	888	19.2	4.0	25.8	5.2
Maize silage	889	9.0	2.5	20.7	29.4
Alfalfa	663	17.5	2.0	31.2	5.0
Wheat straw	425	4.1	1.2	41.8	0.0
Potato pulp	1031	8.1	0.2	20.6	20.2
Brewer’s grains	948	25.8	10.3	18.0	4.0
Sugar beet pulp	1060	8.4	0.6	19.5	4.5
Potato peelings	1112	11.5	0.9	3.9	54.6
Molasses	516	4.9	0.1	0.1	45.1
Grass meal	837	17.7	3.8	21.1	10.3
Soybean meal	1015	46.4	1.6	5.5	9.5
Rape seed meal	879	34.4	3.2	12.1	9.1
Maize gluten feed	975	20.3	4.0	7.3	14.6

^1^ According to Dutch standards [5]; ^2^ 1000 VEM = 6900 kJ/kg Net Energy of Lactation (NEL); ^3^ the amount of protein of animal feed or specific food, depends on the nitrogen content of the food proteins; ^4^ free lipid content; ^5^ residue cellulose material; ^6^ Carbohydrate content.

**Table 2 animals-11-02293-t002:** Main sources used in pig diets and the average nutrient composition [5].

		Nutrients (%)				
Resource	Energy Value (EV) ^1^	Crude Protein	Crude Fat	Crude Cellulose	Starch	Non-Starch Polysaccharides
Cereals (wheat, barley, rye, maize)	1.04–1.23	8–11	1–5	2–5	50–60	11–21
Wheat middlings	0.75	15	4	9	18	39
Tapioca	1.08	2	1	5	62	17
Peas	1.08	21	2	5	39	19
Potato protein	1.04	78	3	1	1	6
Soybean meal	0.94	46	3	4	1	22
Sunflower seed meal	0.71	38	3	15	1	37
Rapeseed meal	0.71	36	3	12	1	34
Lupinen	0.97	35	6	14	2	41
Palm kernel expeller	0.87	15	8	20	0	62
Whey powder	1.14	25	5	0	0	3
Sugarbeet pulp	1.04	9	1	17	1	62
Fat (animal/plant)	3.75	0	99	0	0	0
Sugercane molasses	0.74	4	0	0	0	15

^1^ Energy value (EV) is net energy (NE) of a resource divided by 8.8 MJ.

**Table 3 animals-11-02293-t003:** DEKALB nutrient requirements for pullets and laying hens (personal information from Hendrix Genetics, 2009) in a moderate climate (18–24 °C).

	Starter	Grower	Pullet	Pre-Lay	Early Lay	Lay
Period (days)	28	42	42	~14	~70	364–504
Metabolizable energy (MJ/kg)	12.4	12.0	11.5	11.5		
Feed intake (g/day)					105–125	105–125
Crude protein (%)	20.5	19.0	16.0	16.8	16.2–18.7	15.4–17.9
AAs (%)						
Methionine	0.48	0.41	0.30	0.38	0.37–0.44	0.35–0.41
Methionine + cystine	0.78	0.66	0.53	0.60	0.59–0.70	0.55–0.66
Lysine	1.00	0.85	0.64	0.71	0.68–0.81	0.64–0.77
Threonine	0.67	0.57	0.43	0.48	0.47–0.57	0.45–0.53
Tryptophan	0.186	0.166	0.145	0.155	0.15–0.18	0.14–0.17

**Table 4 animals-11-02293-t004:** Estimated requirements for standardized ileal digestible (SID) amino acids for laying hens to reach maximum production results (egg production and feed conversion). Requirements are based on a laying hen with an egg production rate of 95% producing an egg with a weight of 60 g.

SID Amino Acid ^1^	mg SID-AA per g Egg Mass	Dietary Content (g/kg) ^2^	Ratio to Lysine (%)
Lysine	13.9	6.9–6.2	100
Methionine	7.6	3.8–3.4	55
Methionine + cysteine	12.3	6.1–5.5	88
Threonine	9.7	4.8–4.3	70
Tryptophan	3.0	1.5–1.3	22
Valine	10.8	5.4–4.8	78
Isoleucine ^3^	11.1	5.5–4.9	80

^1^ Only the requirements for the first limiting amino acid requirements are presented; ^2^ based on a daily feed intake of, respectively, 115 and 128 g per laying hen and corresponding with body weights of 1.5 and 2.0 kg; ^3^ the isoleucine requirement also depends on the dietary leucine content. A minimum SID-isoleucine:SID-leucine ratio of 45% is required for optimal performance.

**Table 5 animals-11-02293-t005:** Estimated requirements for standardized ileal digestible (SID) amino acids for broilers fed ad libitum (in g/kg of feed) ^1^.

	Maximum Growth				Minimum Feed Conversion			
Age (weeks)	1	2	3–4	>4	1	2	3–4	>4
Lysine (g/kg)	12.0	11.0	10.2	9.7	12.3	11.4	10.7	10.3
Glycine + serine ^2^	15.0	15.0			15.0	15.0		
MEbr (MJ/kg)	11.9	12.2	12.3	12.6	11.9	12.2	12.3	12.6
(kcal/kg)	2850	2925	2950	3000	2850	2925	2950	3000
**Amino acid (as a ratio to lysine in % and ± st. dev.)**								
Methionine + cysteine	73 ± 7.4				74 ± 8.3			
Methionine ^3^	40				41			
Threonine	64 ± 6.7				62 ± 7.4			
Tryptophan	15 ± 1.2				13 ± 1.7			
Valine	77 ± 5.0				73 ± 5.2			
Isoleucine	60 ± 4.5				63 ± 7.3			
Arginine	107 ± 1.8				112 ± 8.3			
Leucine	110				110			

^1^ Applying a feeding schedule where the birds are fed restrictedly—at least during a certain period—may lead to different requirements. The dietary SID amino acid contents are attuned to the dietary energy content. In case diets with deviating MEbr contents are used, it is also necessary to recalculate the required dietary SID-lysine contents as is described in CVB documentation report no. 62 using formula F.5 for maximum body weight gain and formula F.9 for minimum feed conversion ratio; ^2^ the synthesis capacity of the chick is likely only insufficient from 0 to 2 weeks of age to fully cover the requirements; therefore, the feed should contain a certain minimal amount of these amino acids in this period; ^3^ the methionine requirement is estimated to be 55% of the estimated methionine and cysteine requirement.

**Table 6 animals-11-02293-t006:** Present situation of percentage HEP in the diet of livestock.

Species	Dietary HEP (%)
Dairy cattle	4.5
Pigs	25
Laying hens	30
Broilers	21–24

**Table 7 animals-11-02293-t007:** Protein content of various plant protein sources [7].

Protein Source	Protein Content (%)
Oil seeds—soybean	40
Oil seeds—rapeseed	25
Oil seeds—sunflower	23
Legumes (pulses)—peas/beans/lupine	17–35
Legumes (forage)—lucerne	19
Cereals—oat	12–15
Pseudo cereals—quinoa	12–18
Leaves—grass	12–25
Leaves—(e.g., sugar beet leaves)	12
Macro algae—seaweed	10–30
Micro algae	25–50
Duckweed	35–45
Wheat (as reference)	11

**Table 8 animals-11-02293-t008:** Summary of the by-product protein sources used in livestock to replace SBM and maximum inclusion levels without impaired animal performance. For details and references see the text below.

Sources	Livestock Species	Maximum Dietary Level without Impaired Animal Performance and Productivity
*Camelina sativa* (CS)	Ruminants	
	Dairy cattle	Up to 15%, 4% recommended as optimum
Field peas	Dairy cattle	No recommendation
Lupine, pea seeds mixtures	Dairy cattle	75%
Urea	Dairy cattle	2.1%
Canola meal/RSM	Dairy cattle	5–20%, depending on processing method
SFM	Dairy cattle	10%
Rape seed expeller	Pigs	
*Brassica juncea*	Pigs	No recommendation
SFM, rapeseed cake, field peas mixtures	Pigs	
Rapeseed, cotton, sunflower products	Pigs	No recommendation
Rapeseed press cake, legume seeds mixtures	Pigs	
RSM	Pigs	Up to 21%
Guar		Poultry
SFM	Layers	No recommendation
*Lupinus angustifolius*	Layers	Up to 20%
While flowered semi leafless green spring peas	Layers	300 g/kg
Faba beans	Layers	50 g/kg
Field peas	Broilers	No recommendation
Field peas, faba beans mixtures	Broilers	Potentially OK
Alfalfa peas/meal, faba beans, black soldier fly larva mixtures	Broilers	Potentially OK

**Table 9 animals-11-02293-t009:** Estimated percentage HEP of sources used in livestock diets (Vermeij, personal communication).

Source	Human-Edible Proportion (%)
Cereals and cereal by-products	25
Skimmed milk powder	100
Legumes	25
Oilseeds (soybeans, rapeseed, sunflower)	20
Vegetable oils	75

**Table 10 animals-11-02293-t010:** Heritability of different traits related to feed efficiency in pigs, laying hens, broilers, and dairy cattle.

Trait	Pigs	Laying Hens	Broilers	Dairy Cattle
RFI	0.11–0.38 [73,74,75]	0.21–0.47 [76,77]	0.21–0.49 [78,79,80,81]	0.22–0.40 [82,83]
DFI	0.19–0.56 [73,74,75]			
FI		0.20–0.46 [76,77]	0.35–0.48 [78,79,81]	
ADG	0.24–0.54 [73,74,75,84]			
FE			0.29 [78]	
BF	0.54–0.67 [74,75,84]			
LGR	0.39–0.46 [73,84]			
GR	0.26–0.32 [85]			
FCR	0.28–0.32 [73,74]	0.13–0.19 [77]	0.12–0.49 [78,79,80]	0.17 [86]
BC	0.13–0.19 [87]			
AMEn			0.37–0.38 [81]	
DL			0.47 [81]	
DS			0.37 [81]	
DP			0.33 [81]	
MUN				0.14 [88]
EB				0.20 [88]
DMI				0.29–0.32 [82]

RFI: residual feed intake, DFI: daily feed intake, FI: feed intake, ADG: average daily gain, FE: feed efficiency, BF: back fat, LGR: lean growth rate, GR: growth rate, FCR: feed conversion ratio, BC: body confirmation; AMEn: apparent metabolized energy, DL: digestibility of lipids, DS: digestibility of starch, DP: digestibility of proteins measured in the manure, MUN: milk urea nitrogen, EB: energy balance, DMI: dry matter intake.

## Data Availability

All data were taken from published literature and the internet and are freely available.

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
