# Peer review of "Adaptation of Livestock to New Diets Using Feed Components without Competition with Human Edible Protein Sources—A Review of the Possibilities and Recommendations"

_animals, 2021, doi:10.3390/ani11082293_

Round 1

Reviewer 1 Report

The paper presented for review, "Adaptation of livestock to new diets using feed components without competition with human edible protein sources - a view of the possibilities and recommendation" raises a very important problem from the point of view of the economics of animal husbandry and the use of our planet's natural resources.
The authors presented with great care the current state of knowledge on the possibility of using various sources of protein in the nutrition of farm animals. In the work, they showed possible sources of obtaining nutrients and their impact on the physiology and productivity of animals.
As is known, the way of nutrition has a large impact on the nutritional value of raw materials obtained from animals used for human consumption (which is the main goal of livestock breeding) and the sensory quality. It would be good to expand the work to discuss the impact of the use of human inedible protein components (HIP) in animal nutrition on the sensory characteristics of the raw materials obtained for human consumption.

Author Response

The reviewer asked for the sensory aspects of new feed sources. This is an interesting, but difficult to answer question. The knowledge on this subject seems to be scarce, or not existing. Therefore, we have added the following discussion: Two other aspects of using HIP require further investigation: (1) sensory aspects of HIP-containing feed for the appreciation of the livestock related to willingness to eat and appetite, and (2) sensory aspects of the animal-derived human food products due to the use of HIP in the animal feed. Presently, knowledge of this is lacking, and positive associations between livestock appetite for HIP-containing feed and human sensory evaluation of the derived food products could stimulate the use of HIP in livestock feed.

Reviewer 2 Report

Dear authors of the manuscript,

congratulation on your work, which is well understandably described and is relevant for animal production. In my opinion, it would be important to implement the outcome of the review into the abstract. It would be possible to incorporate to abstract which possible source of human inedible protein components may be used and their advantages as well as disadvantages in the short form.  Just a small remark to p.3 "The Netherlands" the article should be changed to lower case letter. 

Author Response

There is some indication in the abstract of what is in the review. Naming all the numerous new sources would make the abstract too long, and less readable. We hope the reviewer can follow us in this.

Being Dutch myself I know that The Netherlands is the correct spelling of my country name. It will take too long to explain how this name became this way in the history of our country (in short: from the merging of 11 provinces in several intermitting stages....). The name needs 2 capitals. Sorry for this.

Reviewer 3 Report

There is a extensive review of current data which is integrated into the general theme of replacing HEP. The key component of the manuscript is the 4 recommendations which identify key future research objectives.

There are a couple of points worthy of consideration

P 14:’ (>1.7 kg of 3.5% fat corrected milk divided by kg of DMI [56, 57]’ add (>1.7 kg of 3.5% fat corrected milk divided by kg of DMI [56, 57])

P 17: ‘Feeding levels can affect both apparent and true total tract in piglets [100].’ ‘total tract feed digestibility in piglets. Clarity needed

Author Response

P 14:’ (>1.7 kg of 3.5% fat corrected milk divided by kg of DMI [56, 57]’ add (>1.7 kg of 3.5% fat corrected milk divided by kg of DMI [56, 57]) Done

P 17: ‘Feeding levels can affect both apparent and true total tract in piglets [100].’ ‘total tract feed digestibility in piglets. Clarity needed We have added information from the reference: These authors investigated ileal and total tract digestibility in relation to feed intake levels and found that estimates for true digestibility were lower compared to their ap-parent values. They concluded that their results suggested that feed intake level can affect both apparent and true total tract nutrient digestibility in piglets.

Reviewer 4 Report

The manuscript "Adaptation of livestock to new diets using feed components without competition with human edible protein sources – a re-view of the possibilities and recommendations" it is interesting review about one of the hot topic nowadays. The manuscript is well written, in some case there large part without a citation, with some data or in some tables which need to be corrected (see in annex, I give some examples). Other question, is in the alternative protein sources, there is a lack of discussion of each one using data from literature. 

This manuscript can be more valued with some more intrinsic discussion, such epigenetics and its impacts. And not focus in one country or world area, due to be a global question by the SDGs by the UN.

In conclusion, the manuscript can be improved to have more impact in the actual society about a crucial topic.

And The citation style needs to be a unique style, not three (see the annex).

nd the manuscript structure is also strange and needs several restructuration.

Author Response

The manuscript "Adaptation of livestock to new diets using feed components without competition with human edible protein sources – a re-view of the possibilities and recommendations" it is interesting review about one of the hot topic nowadays. The manuscript is well written, in some case there large part without a citation, with some data or in some tables which need to be corrected (see in annex, I give some examples). We have taken the required action on all questions in the annex. Thank you for being so detailed in your comments! Please note that in the case of table 9 we have included text with more details and references before the Table. In the header of the Table we refer to these references. We did so because you made us aware of the fact that the information in the Table was more like a summary rather than be specific for a review. Now the information is more complete. Thank you for making us aware of this. Other question, is in the alternative protein sources, there is a lack of discussion of each one using data from literature. We have added 42 new references to the text. We hope that the reviewer is satisfied on this point. Of course all new references were formatted in the format of the journal and the numbering has been adapted accordingly throughout the text.

This manuscript can be more valued with some more intrinsic discussion, such epigenetics and its impacts. Done. We have included the text: The environment can also influence the expression of the genotype via epigenetic modifications such as DNA methylation and histone modifications to induce mitotically heritable changes in gene expression without altering DNA sequence [103, 104]. In gene promoters methylation generally leads to decreased levels of transcription via alteration of transcription factor binding or changed chromatin conformation [105, 106] while gene body methylation is generally associated with increased levels of transcription [107-109]. The methylation level of genes and genomes has been shown to be significantly associated with complex and disease traits [110, 111]. And not focus in one country or world area, due to be a global question by the SDGs by the UN. Included now.

In conclusion, the manuscript can be improved to have more impact in the actual society about a crucial topic. Thank you. We hope the reviewer finds the revised manuscript improved for this too.

And The citation style needs to be a unique style, not three (see the annex). Done. Thank you for making us aware of this mistake.

nd the manuscript structure is also strange and needs several restructuration. We have included more information and references (42) and improved the manuscript in several ways. We hope that the reviewer is satisfied.

Round 2

Reviewer 4 Report

Thank you for your revisions, all my questions were addressed.

This time, there only some minor revisions (mainly references and text alocation):
Table 1 footnote: reference citation wrong

Table 2: the citation need to be associated to references and not in the the footnote (and table 1 also can be done that way). see example in a recent journal review, such as: https://doi.org/10.3390/ani11072049  or the table 3 (uniformity in the citation in the tables).

Table 4: wrong citation... (in this case, can be (Hendrix Genetics (citation number), without year)

Table 9: the discussion about the data inserted in the table, is below the table and not above (thus the reder see first the table and after the discussion about the data), in that case you can maintain the "For details and references see the text below." If not, each line of data needs a reference (plus column in the table about the reference cited).

Author Response

Thank you for your revisions, all my questions were addressed.

This time, there only some minor revisions (mainly references and text alocation):

Table 1 footnote: reference citation wrong corrected: [5]

Table 2: the citation need to be associated to references and not in the the footnote (and table 1 also can be done that way). see example in a recent journal review, such as: https://doi.org/10.3390/ani11072049  or the table 3 (uniformity in the citation in the tables). Please note that in all three Tables 1, 2, and 3 all data were extracted from the reference. Therefore, it is better in our view to provide the reference once in a footnote, not associated with each data in the Table. What we have done: inserted the website as a new reference [10], and changed the reference numbers from here till the end of the manuscript accordingly.

Table 4: wrong citation... (in this case, can be (Hendrix Genetics (citation number), without year) This is difficult: this is not a real reference that can be found on the internet or in a book, or somewhere else. This is information from a company, which was in an annual report. However, we feel what the reviewer means. Therefore, we have changed the text and added: information from Hendrix Genetics. We hope that this satisfies the reviewer.

Table 9: the discussion about the data inserted in the table, is below the table and not above (thus the reder see first the table and after the discussion about the data), in that case you can maintain the "For details and references see the text below." If not, each line of data needs a reference (plus column in the table about the reference cited). Adding a column in the Table would be very difficult since there is not much space available. Therefore, we have changed the text as the reviewer suggested: see the text below. We will ask the editor is this is correct, or that the text is above the Table (as it is now).